# pFOE or pFTOE as an Early Marker for Impaired Peripheral Microcirculation in Neonates

**DOI:** 10.3390/children9060898

**Published:** 2022-06-16

**Authors:** Nina Hoeller, Christina H. Wolfsberger, Daniel Pfurtscheller, Corinna Binder-Heschl, Bernhard Schwaberger, Berndt Urlesberger, Gerhard Pichler

**Affiliations:** 1Division of Neonatology, Department of Paediatrics and Adolescent Medicine, Medical University of Graz, 8036 Graz, Austria; christina.wolfsberger@medunigraz.at (C.H.W.); daniel.pfurtscheller@medunigraz.at (D.P.); corinna.binder@medunigraz.at (C.B.-H.); bernhard.schwaberger@medunigraz.at (B.S.); berndt.urlesberger@medunigraz.at (B.U.); gerhard.pichler@medunigraz.at (G.P.); 2Research Unit for Neonatal Micro- and Macrocirculation, Department of Paediatrics and Adolescent Medicine, Medical University of Graz, 8036 Graz, Austria

**Keywords:** peripheral muscle, fractional oxygen extraction, near-infrared spectroscopy, neonates, microcirculation

## Abstract

Background: Peripheral-muscle-fractional-oxygen-extraction (pFOE) and peripheral-muscle-fractional-tissue-oxygen-extraction (pFTOE) are often equated, since both parameters are measured with near-infrared-spectroscopy (NIRS) and estimate oxygen extraction in the tissue. The aim was to investigate the comparability of both parameters and their potential regarding detection of impaired microcirculation. Methods: Term and preterm neonates with NIRS measurements of upper (UE) and lower extremities (LE) were included. pFOE was calculated out of peripheral-muscle-mixed-venous-saturation (pSvO_2_), measured with NIRS and venous occlusion, and arterial oxygen saturation (SpO_2_). pFTOE was calculated out of peripheral-muscle-tissue-oxygen-saturation and SpO_2_. Both parameters were compared using Wilcoxon-Signed-Rank-test and Bland–Altman plots. Results: 341 NIRS measurements were included. pFOE was significantly higher than pFTOE in both locations. Bland–Altman plots revealed limited comparability, especially with increasing oxygen extraction with higher values of pFOE compared to pFTOE. Conclusion: The higher pFOE compared to pFTOE suggests a higher potential of pFOE to detect impaired microcirculation, especially when oxygen extraction is elevated.

## 1. Introduction

Peripheral muscle near-infrared-spectroscopy (NIRS) measurements are of increasing interest, especially for early detection of impaired microcirculation in neonates, while other routine vital parameters, such as heart rate, arterial oxygen saturation (SpO_2_), measured by pulse oximetry, or arterial blood pressure, remain within normal range [1,2]. NIRS enables non-invasive measurement of peripheral muscle oxygenation and perfusion. Peripheral muscle tissue oxygen saturation, measured with NIRS, reflects mean haemoglobin oxygenation in venous (70%), capillary (20%) and arteriolar (10%) compartments [3]. When combining peripheral muscle NIRS measurements with venous occlusions, additional information about the measured tissue’s peripheral oxygenation and perfusion status can be gained. Peripheral-muscle-mixed-venous-saturation (pSvO_2_) reflects changes in the venous compartment, when using NIRS combined with venous occlusions. Therefore, the measured pSvO_2_ values are lower than the measured peripheral muscle tissue oxygen saturation values [4]. Furthermore, peripheral-muscle-fractional-oxygen-extraction (pFOE) can be calculated using pSvO_2_ combined with SpO_2_ [3,4]. pFOE provides essential information on both, oxygen consumption and oxygen delivery, depending on the oxygen content of the blood and perfusion [5]. Pichler et al. [1] and Mileder et al. [6] demonstrated that neonates with inflammation have higher pFOE values compared to a healthy control group. The increase of pFOE seems to be a compensation mechanism for the local reduction of peripheral blood flow in the early stages of microcirculation impairment. However, in the literature pFOE is commonly equated with peripheral-muscle-fractional-tissue-oxygen-extraction (pFTOE), which is calculated using peripheral muscle tissue oxygen saturation, also measured with NIRS, instead of pSvO_2_ in combination with SpO_2_ [7]. 

The aim of the present study was to investigate the comparability of pFOE and pFTOE, to find out which of these two parameters might have the higher potential to serve as an early marker for impaired peripheral microcirculation in neonates. 

## 2. Materials and Methods

### 2.1. Patients

In this study, post hoc analyses of secondary outcome parameters obtained from prospective observational studies were performed. Term and preterm neonates with peripheral muscle NIRS measurements with venous occlusion on the upper and/or lower extremities, which were conducted from 2005 to 2016 at the Division of Neonatology of the Department of Paediatrics and Adolescent Medicine, Medical University of Graz, were eligible for this study. NIRS measurements were part of prospective studies, which were approved by the ethics committee of the Medical University of Graz (EK numbers: 14-052 ex 03/04, 19-291 ex 07/08, 21-149 ex 09/10, 23-402 ex 10/11, and 25-237 ex 12/13). In each neonate, parental informed consent was obtained before starting the measurements. Furthermore, we obtained demographic and clinical data, including gestational age, birth weight, sex, umbilical artery pH, and Apgar scores from clinical records for each neonate. Only neonates without signs of inflammation, asphyxia, or congenital malformations were included for the present analyses.

### 2.2. NIRS

NIRS measurements were carried out with the NIRO 300 (until September 2013) and the NIRO 200 NX (since October 2013; Hamamatsu Photonics, Hamamatsu City, Japan). The optodes were placed on the right forearm or the left lateral calf, with an interoptode distance of either 3.0 cm in neonates > 1500 g or 2.0 cm in neonates < 1500 g. A differential path length factor of 5.51 was used [8]. The sampling rate was 2/s. 

Both NIRO devices use the spatially resolved method, which enables the non-invasive continuous measurement of the peripheral-muscle-tissue-oxygenation-index (pTOI) and changes in the concentration of oxygenated haemoglobin (HbO_2_) and deoxygenated haemoglobin (Hb). Derived from the changes in HbO_2_ and Hb, the changes in the concentration of total haemoglobin (HbT) can be calculated. HbO_2_, Hb and HbT were calculated in μmol units.

### 2.3. Venous Occlusion

Venous occlusion was performed using a pneumatic cuff placed around the right upper arm or the left thigh, respectively. Venous occlusion causes an increase in forearm and calf blood volume by undisturbed arterial (in)flow and interrupted venous (out)flow. Thus, changes in HbO_2_, Hb and HbT during venous occlusion are caused only by arterial inflow and oxygen consumption of tissue [9].

### 2.4. Protocol

Measurements were performed under standardized conditions during undisturbed daytime sleep after feeding [10]. The neonates were lying in a supine position, head tilted up 10°. NIRS optodes were positioned on the right forearm or left lateral calf just above the level of mid sternum. Heart rate and SpO_2_ were measured by pulse oximetry using the ipsilateral arm or foot. After positioning of the NIRS optodes, the pneumatic cuff and the pulse oximetry sensors, a calm-down period was introduced until there was at least a 3-min resting period without any body movements. Afterwards, arterial blood pressure was measured oscillometrically with the pneumatic cuff on the right upper arm or the left thigh. After another resting period of 1 min, the pneumatic cuff was inflated within 0.5–1 s to a pressure below the diastolic arterial pressure and above the venous pressure (i.e., 20–30 mmHg). The cuff remained inflated for 20 s and NIRS data were recorded. This procedure was repeated at least five times with a resting period of at least 40 s between inflations and/or until one measurement passed the first quality criterion [4]. The changes in pTOI, HbO_2_, Hb and HbT during these venous occlusions were recorded in a polygraphic computer system (alpha-trace digitalMM, B.E.S.T. Medical Systems, Vienna, Austria).

### 2.5. Calculations

pSvO_2_ was calculated as the ratio of changes in HbO_2_ (ΔHbO_2_) to changes in HbT (ΔHbT): ΔHbO_2_/ΔHbT. pTOI was calculated using the following equation: HbO_2_/HbT. pFOE and pFTOE were calculated using the equations (SpO_2_-pSvO_2_)/SpO_2_ and (SpO_2_ − pTOI)/SpO_2_, respectively.

### 2.6. Statistical Analysis

Only data from measurements passing both recently published quality criteria were included for further analysis [4]. Data are presented as mean and standard deviation (SD) or median and range for continuous data and absolute and relative frequency for categorical data, respectively. Differences between pFOE and pFTOE were analyzed using the Wilcoxon Signed-Rank test for non-normally distributed data. A *p*-value of <0.05 was considered statistical significant. Furthermore, differences between pFOE and pFTOE were graphically displayed using Bland Altman plots. Statistical analyses were performed using SPSS 24.0 (SPSS, Chicago, IL, USA).

## 3. Results

A total number of 341 peripheral muscle NIRS measurements (81 upper extremities [UE], 260 lower extremities [LE]) in term and preterm neonates were included. Demographic, clinical and monitoring data of included neonates are presented in Table 1.

The results for the measured mean values were pTOI 69% ± 6% for UE and 70% ± 6% for LE and pSvO_2_ 66% ± 6% for UE and 67% ± 7% for LE. Comparing the mean values of pFOE and pFTOE, there were significant differences between these two parameters for both locations, UE and LE. UE: pFOE 0.31 ± 0.06 vs. pFTOE 0.28 ± 0.06; *p* < 0.001; LE: pFOE 0.30 ± 0.07 vs. pFTOE 0.27 ± 0.06; *p* < 0.001. In both locations, mean pFOE values were higher, compared to pFTOE values. Graphical visualization of the two parameters of the upper and lower extremities using Bland–Altman plots also showed limited comparability (Figure 1) for both upper and lower extremities measurements. Furthermore, with increasing oxygen extraction, the mismatch between pFOE and pFTOE became more apparent, with pFOE having higher values compared to pFTOE. 

## 4. Discussion

During the last years, NIRS has become an emerging method for non-invasive measurement of tissue oxygenation and perfusion. Nevertheless, Hunter et al. [11] recently demonstrated that there is still limited use in the clinical routine due to insufficient evidence of the advantages of NIRS. In order to reduce interpretation pitfalls, this study concentrated on the comparability of two frequently used NIRS parameters, pFOE and pFTOE, in term and preterm neonates. Comparability of these two parameters was examined for upper and lower extremities, respectively. Comparison of mean values of pFOE and pFTOE as well as Bland–Altman plots revealed limited agreement between these two absolute parameters, in both, UE and LE. Naulaers et al. [12] compared FOE and FTOE of cerebral NIRS measurements in newborn piglets. They found a good correlation between these two parameters and concluded that FTOE can be seen as a trend parameter for FOE. In our study, pFOE and pFTOE showed the same trend, but mean values were significantly different, with pFOE values being significantly higher compared to pFTOE values. These findings are not surprising, since pFOE is calculated using SpO_2_ and pSvO_2_, reflecting the extraction from the arterial to venous compartment, while pFTOE is calculated using SpO_2_ and pTOI, reflecting a mixed saturation compartment including venules, arterioles and capillaries. This implies higher pTOI values than pSvO_2_ values, and as a consequence pFOE was higher than pFTOE [3]. These findings are in accordance with Pichler et al. [7], where pFOE values were also higher than pFTOE values of the lower extremities in term and preterm neonates (0.30 ± 0.07 vs. 0.26 ± 0.07). Additionally and even more interesting, our analyses showed that with increasing oxygen extraction, the differences between the two parameters are even more pronounced, highlighted by an increasingly scattered pattern in the Bland–Altman plots in those regions (Figure 1). An increase in pFOE and pFTOE indicates a status of reduced peripheral blood flow [1,6], which can be assumed as an early sign for compensated shock. A further reduction of oxygen delivery beyond “the critical O_2_ point” leads to a dramatic increase in oxygen extraction to meet the metabolic needs of the tissue [13]. Due to the pronounced discrepancy between the two parameters with increasing pFOE values, it seems that in the case of impaired microcirculation, pFTOE might underestimate the true magnitude of the ongoing processes of the microcirculation. The reason why pFTOE is commonly used in studies rather than pFOE lies in the measurement/calculation effort of the two parameters. While pFTOE is calculated using SpO_2_ and pTOI, which are both measured non-invasively and instantaneously via pulse oximetry and NIRS [7], pFOE requires the measurement of pSvO_2_, which can either be estimated invasively by blood samples from central venous catheters using co-oximetry or non-invasively with NIRS in combination with venous occlusions [5,14]. pSvO_2_ estimation is more complex and more irritating for the neonates, and therefore pFTOE seems to be the simpler and more convenient alternative. Nevertheless, this study implicates that in neonates with risk of microcirculation impairment, either due to inflammation or other reasons for shock, pFTOE may show the same trend, but pFOE might provide more reliably important additional information about the magnitude of disturbances. As a future perspective, Mintzer at al. [15] suggested implementing pulse oximetry data to NIRS devices, which might allow real-time tissue-specific FTOE trending.

## 5. Conclusions

Summing up, pFOE and pFTOE, both parameters of peripheral muscle oxygen extraction measured with NIRS, cannot be equated in term and preterm neonates. Although these two parameters show some comparability and the same trend, the values are significantly different with pFOE values being higher than pFTOE values. Furthermore, with increasing oxygen extraction, the differences between the two parameters become even more apparent, indicating that pFTOE might underestimate the ongoing microcirculation disturbances in conditions of compensated shock.

## Figures and Tables

**Figure 1 children-09-00898-f001:**
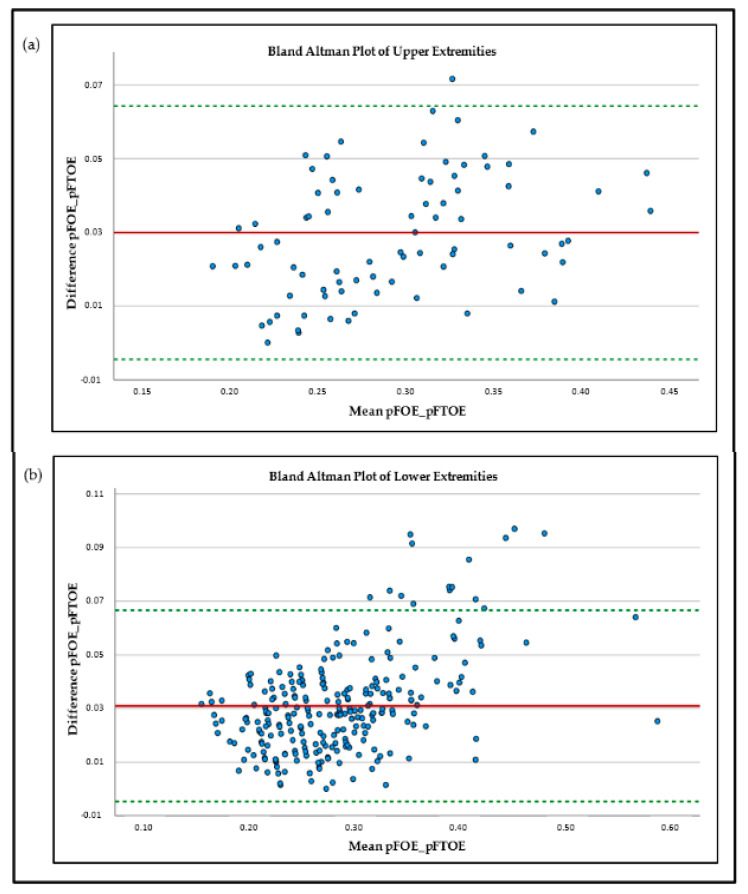
Bland–Altman plots of peripheral-muscle-fractional-oxygen-extraction (pFOE) and peripheral-muscle-fractional-tissue-oxygen-extraction (pFTOE) of upper extremities (**a**) and lower extremities; (**b**) NIRS measurements in neonates.

**Table 1 children-09-00898-t001:** Demographic, clinical and monitoring data of included neonates. Data are presented as mean ± SD or median (range). (* Largest circumference of the extremity was measured with a measuring tape below the sensor placement; ** Diameter and thickness of subcutaneous fat were measured at the largest circumference of the extremity using ultrasound).

	Upper Extremities *n* = 81	Lower Extremities *n* = 260
Female (*n*)	30	99
C-Section (*n*)	42	146
Gestational Age (weeks)	34.6 ± 2.8	34.8 ± 3.2
Birth weight (g)	2241 ± 701	2361 ± 775
Age at initiation of measurement (h)	26 ± 37	72 ± 188
Actual weight (g)	2334 ± 764	2422 ± 797
Umbilical artery pH	7.30 (7.05–7.41)	7.30 (6.99–7.43)
APGAR Minute 1	8 (1–9)	8 (1–10)
APGAR Minute 5	9 (6–10)	9 (2–10)
APGAR Minute 10	10 (8–10)	10 (3–10)
Circumference of lower arm/lower leg (cm) *	8.1 ± 1.1	9.5 ± 2.8
Diameter of lower arm/lower leg (cm) **	2.5 ± 0.5	2.8 ± 0.5
Thickness of subcutaneous fat of lower arm/lower leg (cm) **	0.27 ± 0.09	0.31 ± 0.11
Arterial oxygen saturation (%)	96 ± 3	96 ± 3
Heart rate (bpm)	132 ± 14	132 ± 13
Mean arterial blood pressure (mmHg)	44 ± 8	42 ± 8

## Data Availability

All data generated or analyzed during this study are included in this article. Further enquiries can be directed to the corresponding author.

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
