# Peer review of "pFOE or pFTOE as an Early Marker for Impaired Peripheral Microcirculation in Neonates"

_children, 2022, doi:10.3390/children9060898_

Round 1

Reviewer 1 Report

The early detection of impaired microcirculation in neonates to evaluate tissue’s peripheral oxygenation becomes an important parameter in the daily practice. The peripheral muscle near-infrared-spectroscopy measurements of peripheric circulation are of great interest, along with other routine vital parameters, such as heart rate, arterial oxygen saturation (SpO2), measured by pulse oximetry, or arterial blood pressure.

This study analyzes whether, as previously considered, the peripheral-muscle-fractional-oxygen-extraction and peripheral-muscle-fractional-tissue-oxygen-extraction offer superposable results, since both parameters are measured with near-infrared-spectroscopy (NIRS) and estimate oxygen extraction in the tissue.

As a new and bold hypothesis, the study group comprises healthy neonates. This supposition is thoroughly studied and resulted in significant conclusions. The study should be further continued involving not only healthy neonates.

I suggest to the authors to analyze Line 68 and perhaps replacing ‘from with for’, which would serve better the meaning of this statement.

I also suggest them to consult the following references.

 Mintzer JP, Moore JE. Regional tissue oxygenation monitoring in the neonatal intensive care unit: evidence for clinical strategies and future directions. Pediatr Res. (2019) 86:296–304. doi: 10.1038/s41390-019-0466-9

Hunter CL, Oei JL, Suzuki K, Lui K, Schindler T. Patterns of use of near-infrared spectroscopy in neonatal intensive care units: international usage survey. Acta Paediatr. (2018) 107:1198–204. doi: 10.1111/apa.14271

Reviewer 2 Report

Dear Authors,

congratulations!

best regards

Roksana

Author Response

Thank you very much for your nice comment.

Reviewer 3 Report

The authors describe a comparative study between peripheral-muscle-fractional-oxygen-extraction (pFOE) and peripheral-muscle-fractional-tissue-oxygen-extraction (pFTOE) for detection of impaired microcirculation. The manuscript is well-written and describes the study in adequate detail. The results showed that pFOE and pFTOE are not equatable and there is only limited comparability between the two, with the differences being more pronounced with increasing oxygen extraction, such as venous occlusion performed in this study. 

Some comments: 

1. Title is too wordy and doesn't seem to flow well; can the authors simplify the title to make it brief?

2. The design of the study is well thought out. One concern i have is the use of pneumatic tourniquet to decrease venous return, synonymously with "impaired microcirculation". Using "Impaired microcirculation" seems to suggest tissue hypoxic injury, which is clearly not the case here since the authors appear to have taken precautions to prevent injury. There appears to be no other markers for impaired circulation described here; Please clarify the rationale better. May I kindly suggest using a different phrase such as "decreased venous return" or another such phrase to describe this situation more accurately?

3. Materials and methods: Line 58-60: The authors describe a post-hoc analysis of secondary outcomes. Are the authors referring to a published study? If so, please cite the relevant article here. Is data from this study being published under a different manuscript also? 

4. Table 1: To be thorough, please describe how and where the measurements for limb circumferences, diameters and fat thicknesses were obtained. This may be included in the figure legend. 
